# Cognitive Soil Digital Twin for Monitoring the Soil Ecosystem: A Conceptual Framework

Nikolaos L. Tsakiridis [1,*], Nikiforos Samarinas [1], Eleni Kalopesa [1] and George C. Zalidis [1,2]

1   Laboratory of Remote Sensing, Spectroscopy, and GIS, School of Agriculture,
    Aristotle University of Thessaloniki, 57001 Thermi, Greece; smnikiforos@topo.auth.gr (N.S.);
    kalopesa@agro.auth.gr (E.K.); zalidis@agro.auth.gr (G.C.Z.)
2   Interbalkan Environment Center, 18 Loutron Str., 57200 Lagadas, Greece
*   Correspondence: tsakirin@ece.auth.gr; Tel.: +30-2310-99-1783

**Abstract:** The digital twin concept has found widespread application across diverse industries. Herein, we present a comprehensive conceptual framework for the cognitive soil digital twin, which embodies the intricate physical reality of the soil ecosystem, aiding in its holistic monitoring and comprehension. The digital twin can seamlessly integrate a multitude of sensor data sources, including field Internet of Things sensors, remote sensing data, field measurements, digital cartography, surveys, and other Earth observation datasets. By virtue of its duality, this digital counterpart facilitates data organisation and rigorous analytical exploration, unravelling the intricacies of physical, chemical, and biological soil constituents while discerning their intricate interrelationships and their impact on ecosystem services. Its potential extends beyond mere data representation, acting as a versatile tool for scenario analysis and enabling the visualisation of diverse environmental impacts, including the effects of climate change and transformations in land use or management practices. Beyond academic circles, the digital twin's utility extends to a broad spectrum of stakeholders across the entire quadruple helix, encompassing farmers and agronomists, soil researchers, the agro-industry, and policy-makers. By fostering collaboration among these stakeholders, the digital twin catalyses informed decision-making, underpinned by data-driven insights. Moreover, it acts as a testbed for the development of innovative sensors and monitoring frameworks, in addition to providing a platform that can educate users and the broader public using immersive and innovative visualisation tools, such as augmented reality. This innovative framework underscores the imperative of a holistic approach to soil ecosystem monitoring and management, propelling the soil science discipline into an era of unprecedented data integration and predictive modelling, by harnessing the effects of climate change towards the development of efficient decision-making.

**Keywords:** digital twins; simulation; artificial intelligence; IoT; Copernicus

## 1. Introduction

Monitoring soil health is a paramount endeavour due to its pivotal role in sustaining both agricultural productivity and ecosystem vitality [1,2]. The significance of this practice extends beyond the realms of conventional farming, encompassing broader environmental concerns, global food security, and sustainable land use management [3]; for example, the soil ecosystems serve as the bedrock for sustaining more than 90% of the world's food supply [4] and provide habitat for a substantial 25% of the planet's total biodiversity [5]. By comprehensively assessing soil health, we gain invaluable insights into the intricate web of interactions occurring within the soil matrix, including nutrient cycles, microbial communities, and physical structure. This knowledge, in turn, empowers end-users to make informed decisions to optimise crop yields, conserve natural resources, and minimise environmental impacts.

Nonetheless, the pursuit of accurate soil health assessment is not without its challenges, as traditional methods have proven to be limiting in their scope and precision, whilst there

is not yet a single consensus on the definition of soil quality [6]. Conventional techniques often involve time-consuming and labour-intensive processes that provide only a snapshot of soil conditions, failing to capture the dynamic and spatially heterogeneous nature of soils. Additionally, these methods may lack sensitivity in detecting subtle changes or emerging issues, thus impeding our ability to proactively address soil degradation, contamination, or shifts in fertility patterns. Moreover, traditional approaches can be cost-prohibitive when applied over large areas, hindering widespread adoption among farmers and land managers.

Recognising the threats that hinder soil health and taking into account that over 60% of the European soils are currently in an unhealthy state, the European Union with its "Soil Deal for Europe" aims to establish 100 living labs and lighthouses to lead the transition towards healthy soils by 2030 [7]. Living labs act as real-life test and experimentation environments fostering co-creation and open innovation, involving all actors of the Quadruple Helix Model, i.e., academia, citizens, government, and industry. The success of such an endeavour necessitates the utilisation of capable models that can simulate all of the complex soil processes.

The concept of the digital twin has emerged as a powerful tool in various industries, aiming to facilitate actions such as real-time monitoring, data fusion, simulation, and projections and, overall, provide a holistic understanding of complex systems. It was introduced in 2013, but its definition has since evolved with no widespread consensus on the definition of various types of twins [8,9]. Nonetheless, in principle, a digital twin may be defined as a virtual representation of real-world entities and processes, synchronised at a specified frequency and fidelity [10]. Its purpose is to enable measurements and simulations and act as a test bed for experimentation with a digital representation in order to better understand its physical counterpart, performing monitoring, analysis, and prediction. It requires both efficient and standardised data aggregation capacities and powerful data analytics. Digital twins have been employed, among others, in manufacturing [11], smart cities [12], and oil pipeline risk estimation [13]. The integration of digital twins in agriculture, including soil monitoring, has been performed only at small scales (at the field level) [14–16].

An interesting proposed implementation is the European Space Agency's (ESA) Digital Twin Earth initiative, aiming to create a comprehensive digital replica of our planet, integrating Earth observation data, computational models, and advanced analytics [17,18]. It, thus, provides a means to better understand and manage Earth's complex systems, including the atmosphere, oceans, and land and their interconnections. By harnessing the power of digital twins, ESA can gain valuable insights into climate change, natural disasters, resource management, and other crucial global challenges, helping to visualise, monitor, and forecast natural and human activity on the planet. The model will be able to monitor the health of the planet, perform simulations of Earth's interconnected system with human behaviour, and support the field of sustainable development. It, thus, acts as a vital tool for decision-makers, scientists, and policy-makers, facilitating evidence-based decision-making, informed policy formulation, and proactive planning towards a sustainable future.

In this paper, we postulate that the successful monitoring and analysis of soil dynamics necessitates the integration of diverse technological tools and advances, such as augmented reality (AR) and extended reality (XR) [19], the Internet of Things (IoT), citizen science [20], and artificial intelligence/machine learning (AI/ML) [21]. The convergence of these cutting-edge technologies is crucial for enabling a comprehensive and accurate assessment of soil properties and processes. The integration of IoT devices within the digital twin framework enables real-time monitoring of critical soil parameters, ensuring the continuous acquisition of accurate and up-to-date data. At the same time, data from air- or space-borne sensors are increasingly becoming available in near-real-time; these include multi- or hyper-spectral imaging sensors recording the soil's electromagnetic reflectance in the visible, near-infrared, and thermal infrared spectrum and radar data, whose microwave signals penetrate clouds, haze, and vegetation. This continuous data flow, in conjunction

with AI/ML algorithms, enables advanced analytics to uncover hidden patterns, forecast soil dynamics, and support precision agriculture practices. Only by seamlessly integrating these technological advancements can researchers gain a holistic understanding of soil functioning and develop effective strategies for sustainable land management. Finally, the incorporation of AR and XR technologies provides researchers and stakeholders with immersive visualisation and interactive capabilities, enabling enhanced comprehension, analysis, and decision-making.

The rest of the paper is organised as follows. Section 2 presents the background and current state-of-the-art in digital twin technologies and contemporary tools used in soil monitoring; their limitations are explored, and the proposed advantages of the cognitive soil digital twin are demonstrated. Section 3 outlines the proposed architecture for the digital twin, exploring its components and enabling technologies. The challenges and opportunities of the system are laid out in Section 4, with the conclusions presented in Section 5.

## 2. Background and State-of-the-Art

### 2.1. Digital Twin Technologies

Applications and different manifestations of the digital twin paradigm have been extensively reviewed in the literature in dedicated review papers [9,22–25]. In most cases, the following fundamental characteristics exist across the various implementations of the digital twin paradigm (Figure 1):

- Real-time mapping of a physical entity with high fidelity: The digital twin must be a complete virtual copy of its physical counterpart, with high-precision sensors providing accurate measurements, which, in turn, enable the twin to simulate and predict the different system states.

- Entire lifecycle data management: As the physical counterpart is dynamic, it is necessary to store the entire lifecycle data of the system, enabling functions such as historical state analysis, health analysis, and other data mining and analytical activities. Due to the high volume of such data, distributed data storage architectures must be considered. These elaborate analytical processes rely heavily on the data stored and managed throughout the system's lifecycle, but empower the retrospective examination of the soil's past conditions, which is essential for assessing the performance and health of the system over time.

- Self-evolution: The digital twin should be able to adapt to changes and evolve; changes recorded in the physical counterpart ought to be reflected in the twin, with the data collected in real-time enabling the evolution and maturity of the twin in parallel with the physical counterpart. The self-evolution aspect also concerns the update of the various models and simulations when new data become available.

- Multi-disciplinarity in virtual modelling and simulation: Different disciplines such as computer science, communications and automation and domain-specific knowledge (such as soil science in the case of the soil digital twin) must be fused to provide high-fidelity virtual modelling technologies. In effect, multi-domain data and knowledge coupled with multi-timescale and multi-dimensional information must be combined in order to provide accurate modelling and simulation.

### 2.2. Tools and Techniques for Soil Monitoring

Contemporary soil-monitoring and -mapping techniques encompass a variety of traditional tools that have been refined and augmented with modern technological advancements. Among the commonly employed methods is field sampling, where soil samples are collected at designated locations and depths for laboratory analysis [26]. This technique provides essential information on soil properties such as pH, texture, organic matter content, nutrient levels, and cation exchange capacity [27]. Additionally, soil spectroscopy techniques, such as visible–near-infrared (VNIR) and mid-infrared (MIR) spectroscopy, offer rapid and non-destructive means of assessing soil composition and properties based on the

interaction of electromagnetic radiation with soil constituents [28,29]. Space-borne or airborne remote sensing is also employed to provide detailed maps of soil properties [30–32] or to assess land degradation [33]. Another widely used technique is geophysical methods, including electrical resistivity [34], electromagnetic induction [35], and ground-penetrating radar [36], which enable the estimation of soil properties by measuring variations in electrical or electromagnetic properties. Furthermore, sensor-based technologies, such as soil moisture sensors, temperature probes, and nutrient probes, allow for real-time monitoring of key soil parameters in the field [37,38]. The above notwithstanding, traditional techniques such as soil classification systems (e.g., Soil Taxonomy) [39] and soil mapping using geostatistical approaches [40,41] continue to play a fundamental role in soil characterisation and mapping.

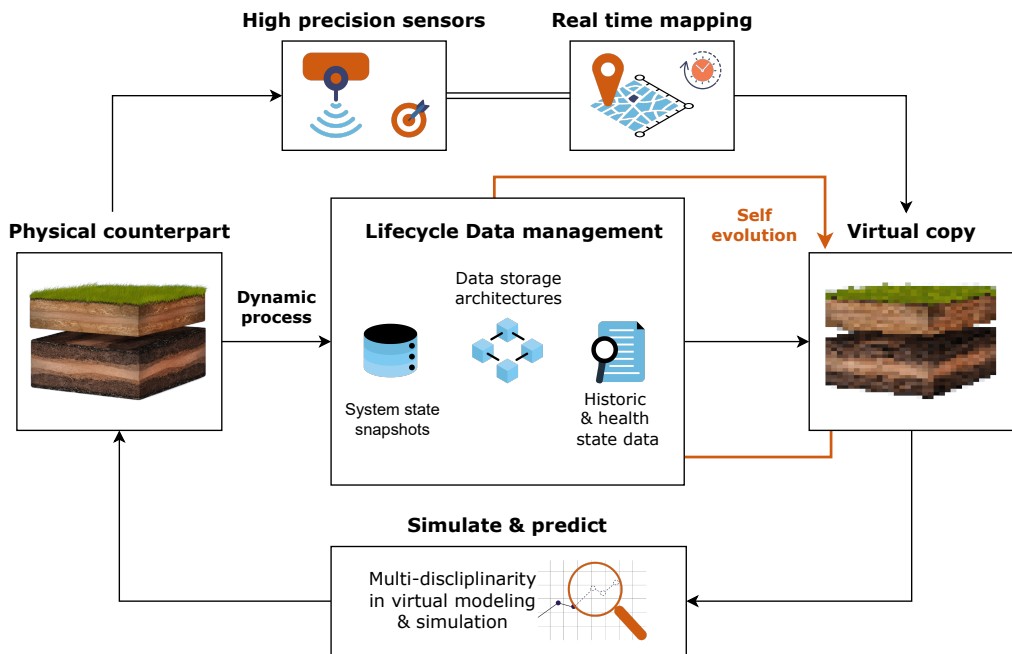

**Figure 1.** The fundamental characteristics of a cognitive soil digital twin.

In addition to the traditional tools mentioned earlier, contemporary soil-monitoring and -mapping techniques encompass advanced tools that provide estimations of carbon sequestration or carbon stock [42–44], the simulation of climatic scenarios or land use changes [45], and other simulations crucial for understanding soil dynamics [46]. One such tool is the RothC model [47] employing carbon models and soil-carbon-mapping techniques, integrating field data, remote sensing data, and computational algorithms to estimate soil organic carbon stocks and changes over time. All these models help assess the potential of soils to sequester carbon and contribute to climate change mitigation strategies. Simulation models, such as ecosystem and agroecosystem models, simulate soil processes under different climatic scenarios, land management practices, and cropping systems. They provide insights into soil–water dynamics, nutrient cycling, crop growth, and carbon fluxes, allowing researchers to project future impacts and optimise land use strategies. Additionally, soil erosion models, such as the Universal Soil Loss Equation (USLE) or the Revised Universal Soil Loss Equation (RUSLE) [48], simulate soil erosion processes and help identify areas prone to erosion, facilitating erosion control and soil conservation efforts. Moreover, hydrological models, such as the Soil and Water Assessment Tool (SWAT) [49], simulate water movement, infiltration, and runoff in landscapes, aiding in water resource management [50] and flood prediction. These advanced tools and simulations play a critical role in assessing the impacts of climate change, land management practices, and policy interventions on soil health, carbon dynamics, water resources, and overall ecosystem sustainability.

### 2.3. Limitations of Contemporary Approaches

The tools usually employed currently are associated with certain limitations. Evidently, the most-notable drawback is the lack of the integration of the multiple input data streams; oftentimes, approaches attempting to monitor the soil focus on one aspect (e.g., certain physicochemical properties) while neglecting the others (e.g., soil biodiversity) that are intertwined. Thus, this shortcoming lies in the scope of simulation and nowcasting capabilities offered by current tools: while they may provide estimations for specific soil properties or processes, they often fall short in simulating the entirety of ecosystem services associated with soils [51]. Furthermore, these models employ different inputs that need to be prepared in advance, and due to the lack of one central repository, it is cumbersome to collect and standardise all necessary Earth observation input layers [52]. Another limitation is the predominantly open-loop control nature of existing approaches. Traditional tools often provide one-time measurements or data collection, lacking the ability to continuously integrate and update information in real-time. Contrarily, if continuous monitoring, analysis, and feedback loops are employed, this can facilitate adaptive management strategies that respond to dynamic soil conditions. Finally, contemporary tools often lack the full utilisation of data analytics and advanced computational techniques. While data may be collected, the potential for extracting valuable insights and patterns through data analytics remains largely untapped.

### 2.4. Advantages of a Cognitive Soil Digital Twin

The following section iterates the various advances that a cognitive soil digital twin offers over the contemporary approaches and how these can be integrated in soil living labs and lighthouses.

#### 2.4.1. Real-Time Monitoring and Data Fusion

Digital twin technologies offer several advantages in terms of real-time monitoring and data fusion within the cognitive soil digital twin framework. Firstly, these technologies enable continuous real-time monitoring of ecosystem parameters, providing up-to-date and dynamic information about soil conditions. By seamlessly integrating disparate data sources, including Earth observation data, sensor networks, and IoT devices, the cognitive soil digital twin fosters a holistic understanding of ecological processes and dynamics. This real-time monitoring capability allows for a synchronisation between the physical entity and the digital twin, which, in turn, enables the timely identification of changes or anomalies in soil properties, enabling proactive decision-making and interventions. Additionally, the cognitive soil digital twin facilitates data fusion, which involves the integration and analysis of multiple data types and sources. By combining data from various sensors, models, and observations, the digital twin can generate a comprehensive and accurate representation of soil characteristics and behaviour. This data-fusion process enables researchers to uncover complex relationships and patterns, leading to deeper insights into soil processes, nutrient cycling, water dynamics, and other critical ecosystem functions. Furthermore, the real-time monitoring and data fusion capabilities of the cognitive soil digital twin support the generation of near-real-time or predictive models, allowing for proactive planning and management. By continuously assimilating new data and updating the models, the digital twin can provide timely and accurate predictions, empowering stakeholders to make informed decisions regarding soil management, crop growth, and environmental sustainability. This closed-loop control approach enables, furthermore, the continuous update of the models simulating the physical counterpart through techniques such as active learning.

#### 2.4.2. Simulation and Scenario Analysis

Computational simulation has emerged as a crucial technique for analysis and decision-making support, demonstrating its significance in diverse sectors such as manufacturing, healthcare facilities, logistics, services, and more [53]. The digital twin paradigm offers

distinct advantages in the realm of simulation and scenario analysis within the soil science domain by providing a powerful platform for simulating a wide range of climatic scenarios and anthropogenic interventions, including agricultural practices such as tillage practices and crop rotation. By leveraging advanced AI/ML algorithms and integrating Earth observation data, these digital twin models possess the capability to predict and simulate ecosystem responses to various scenarios. Through the incorporation of comprehensive data inputs, such as soil properties, climate data, land use patterns, and management practices, digital twin models can generate valuable insights into the potential impacts and outcomes of different scenarios on soil health, crop productivity, and overall ecosystem dynamics. Moreover, the simulation and scenario analysis capacity of the digital twin enables researchers and decision-makers to assess the effectiveness of alternative strategies, optimise resource allocation, and design more-sustainable land management practices. By providing a virtual environment to explore and evaluate different scenarios, the digital twin becomes an invaluable tool for informed decision-making and proactive planning in the realm of soil science research and land management.

### 2.4.3. Policy Support and Environmental Management

The application of digital twin technologies to ecosystem monitoring provides a powerful tool for supporting environmental policies and management strategies. The soil digital twin, for instance, can significantly bolster policy support and environmental management efforts. By offering a comprehensive virtual representation of real-world soil ecosystems, the digital twin facilitates informed decision-making in multiple ways. Firstly, it allows policy-makers to simulate and assess the potential impacts of various policy interventions on soil health and quality. This predictive capability minimises the risk of unintended consequences and enables the formulation of well-informed, targeted policies that align with sustainability objectives.

Moreover, the soil digital twin's ability to integrate real-time data from sensors and satellites ensures that policy recommendations are grounded in accurate and up-to-date information. This dynamic data-driven approach allows for adaptive management strategies that can respond to changing environmental conditions. Through simulation and scenario testing, policy-makers can assess the long-term implications of their decisions and fine-tune strategies for optimal outcomes.

Furthermore, the soil digital twin serves as a platform for collaboration and stakeholder engagement. It enables policy-makers, researchers, and local communities to interact with the virtual representation of the ecosystem, fostering a shared understanding of complex soil interactions and dynamics. This collaborative approach enhances the feasibility of implementing policies by garnering support from diverse stakeholders.

In essence, the soil digital twin acts as a bridge between scientific insights and policy implementation. Its capabilities encompass scenario exploration, data integration, and stakeholder engagement, all of which contribute to evidence-based decision-making and the achievement of the European environmental goals [7]. As a versatile and powerful tool, the soil digital twin empowers policy-makers to navigate the intricate landscape of soil ecosystem management with precision and purpose, thereby propelling sustainable environmental practices forward.

## 3. Proposed Architecture for a Cognitive Soil Digital Twin

The conceptual framework for developing the cognitive soil digital twin is presented in Figure 2. In the subsections that follow, Section 3.1 presents the system components of Figure 2 and describes their constituent parts, as well as their interconnectivity. Following this, Section 3.2 moves on to suggest a potential technological stack for the implementation of these components and of the system as a whole.

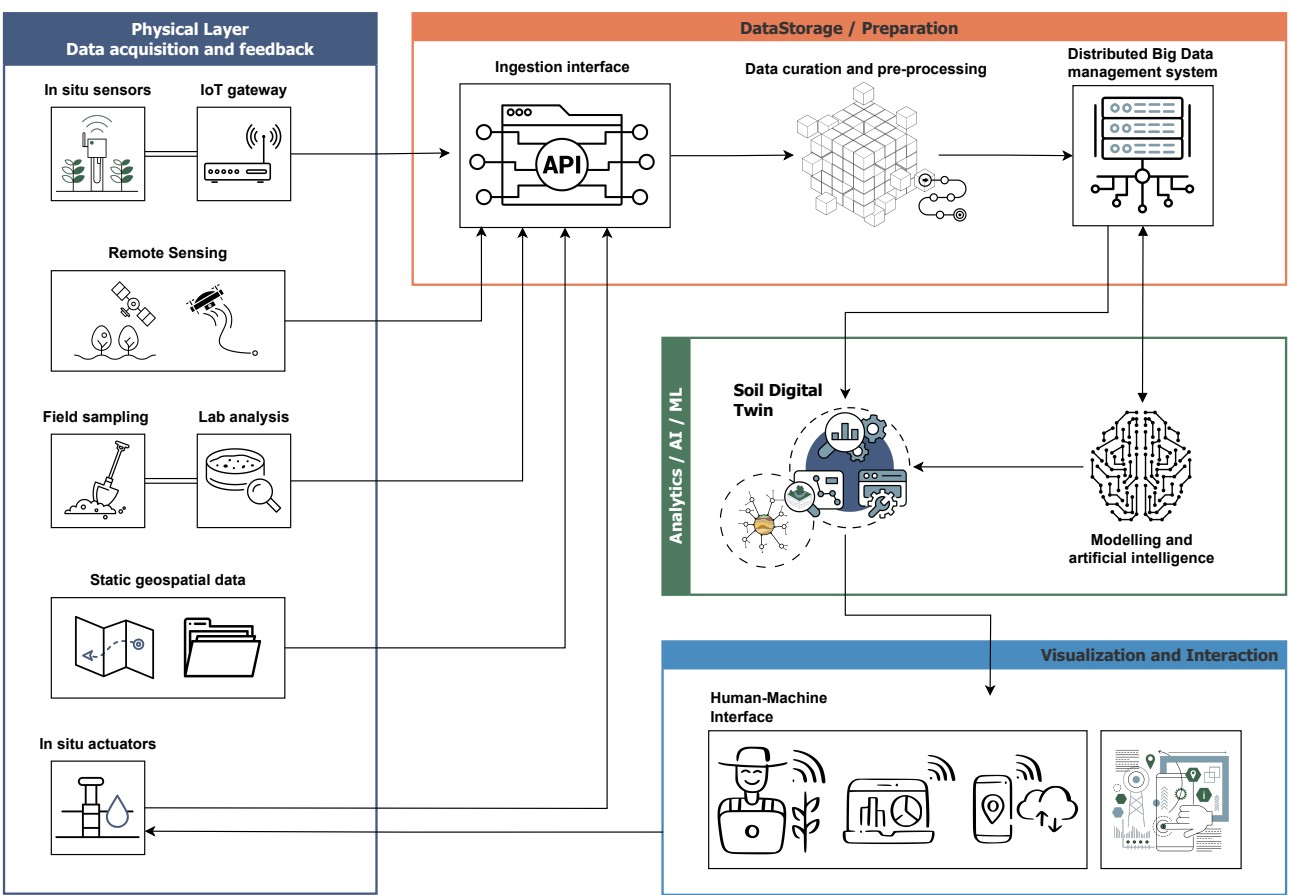

**Figure 2.** The conceptual framework for the proposed cognitive soil digital twin.

*3.1. System Components*

3.1.1. Physical Layer: Data Acquisition and Feedback

A cognitive digital twin is a real-time dynamic mapping of a physical entity and, thus, relies on the integration and fusion of diverse datasets from multiple sources, including Earth observation satellites [54], ground-based sensors and IoT devices [55], and socio-economic data [56]. From a big data point of view, these data streams are high in variety, volume (e.g., space-borne data may be hundreds of GBs depending on the examined period), and velocity [57]. Devices that can be used to apply control and affect the soil such as performing irrigation [58] are also part of this layer; typically, these devices are referred to as actuators [59].

The IoT sensors and actuators are typically connected wirelessly to a gateway using a communication protocol (e.g., LoRA [60,61], Bluetooth, etc.), which, then, relays the collected data over the Internet to the central data repository. These streaming data are continuously updated and ingested in the system at high temporal resolution. The use of actuators that are activated from the gateway, which, in turn, may be activated remotely through the Internet, enables an operator to manually or automatically control some of the parameters of the environment (e.g., irrigation or control of greenhouses). At the same time, the provision of real-time data from the sensors enables the system to have feedback on the actions performed from the actuator or from changes in the environment.

On the other hand, remote sensing data are sparser and come in batches at pre-defined time intervals (e.g., the revisit time of Sentinel-2 is about 4 days) or an on-demand schedule (e.g., UAV flights). The space-borne data may be downloaded directly from the operator (e.g., Copernicus) or other cloud providers that ingest them (e.g., AWS or Google) [62]. An alternative is to use APIs or cloud services to perform processing on distributed cloud back-ends (e.g., the openEO API, Google Earth Engine [63], or the Copernicus Data Space



Ecosystem Services (https://dataspace.copernicus.eu/, accessed on 12 September 2023), where proprietary data must be connected to these services.

Notwithstanding the above, geospatial and other data from miscellaneous sources should also be considered to be integrated into the system. These include climatic data (e.g., ERA5 from the European Centre for Medium-Range Weather Forecasts, ECMWF, or from other local meteo stations) including future projections or data pertaining to land management practices.

### 3.1.2. Data Integration and Fusion

The disparate datasets are ingested, harmonised, and combined to form a comprehensive representation of the Earth's ecosystems. Data may be subject to curation (e.g., outlier removal) [64] or other forms of pre-processing (e.g., the generation of second-order products like the transformation of dielectric permittivity to soil moisture). The data are then stored in a distributed big data management system ideally in an analysis-ready format using pre-specified protocols and standards [65], which offers multifaceted advantages such as scalability, performance, resilience, flexibility, and data consistency.

If data from disparate remote sensing sensors should be integrated, at this stage, techniques such as spatiotemporal fusion [66,67] may be used, which can combine the positive traits of multiple sensors. Moreover, the use of knowledge representation techniques such as geospatial ontologies [68,69] and semantic datacubes [70], which can represent both symbolic and numeric knowledge and share knowledge on the interpretation of these products, can provide a further option to semantically enhance the stored products.

Finally, data from third-party data providers or external APIs should be seamlessly integrated to allow easy callbacks in the modelling stage. This step is, however, contingent on potential limitations. Online APIs oftentimes have limitations like rate limiting (i.e., restrictions on the number of requests within a given time frame), payment requirements for certain features or higher usage thresholds, frequent changes that require upkeep, and sometimes, inadequate documentation, all of which necessitate careful integration planning.

### 3.1.3. Data Analytics: Geospatial Modelling and Simulation

The integration of the data in an analysis-ready format enables the processing of the data streams and the combinations of the various layers of information, either from primary data sources or from generated products via modelling techniques. To make sense of the data, efficient data fusion and analytics must take place.

Geospatial modelling forms a critical component of a cognitive digital twin. It involves the creation of accurate and detailed representations of geographic features, such as land cover, topography, hydrology, and vegetation, but also layers providing spatiotemporal information about the soil's physical, chemical, and biological properties [71,72]. These models, combined with simulation and forecasting capabilities, allow for the analysis of ecosystem behaviour under various conditions and the evaluation of potential interventions [73].

AI and ML algorithms play a pivotal role in a cognitive digital twin [74,75]. They enable the interpretation and analysis of vast amounts of data, pattern recognition, predictive modelling, and decision support. Various soil quality indicators have been analysed using data from remote sensing via AI/ML models [76] with the focus oftentimes placed on croplands [32]. AI/ML techniques facilitate the identification of complex relationships and trends within ecosystem data, improving the accuracy and effectiveness of monitoring and simulation processes.

The above notwithstanding, standard statistical tools and processes may be also fundamental to generate intermediate or final products. For example, multi-temporal analysis [77] to, e.g., generate bare soil reflectance composites [78] is a typical tool employed for monitoring of cropland soils.

### 3.1.4. Visualisation and Interaction

Visualisation tools that may also provide the user with the ability to interact with them or with the digital twin have a two-fold goal:

1.  Enhancing understanding: Visualisation and user interface tools are crucial for digital twins as they translate complex data and simulations into accessible visual representations. These tools provide stakeholders with an intuitive understanding of intricate systems, fostering better decision-making and enabling effective communication across technical and non-technical audiences.
2.  Interactivity and engagement: Visualisation and user interface tools offer interactivity and engagement, allowing users to explore, manipulate, and analyse the digital twin's virtual environment. This hands-on experience not only deepens understanding, but also empowers users to test scenarios, validate hypotheses, and collaborate in real-time, ultimately driving innovation and efficiency in various industries.

Visualisation techniques, including augmented reality (AR), provide a means for researchers and stakeholders to interact with and comprehend complex ecosystem models and data [79]. These visual representations transcend language barriers and technical intricacies, fostering a profound enhancement in the understanding of intricate soil processes and ecosystem dynamics. They can, thus, enhance understanding, aid in decision-making, and facilitate communication between experts and policy-makers by cultivating an environment conducive to effective cross-domain synergies. In the past, AR has been integrated into the digital twin paradigm with much research focused on digital assembly technology [80] and robotics [81]. In addition to providing an immersive and remote-controlled experience of the physical counterpart, virtual reality also empowers human operators to engage with the virtual replicas without causing any disruption to the regular operations of the tangible entities.

In tandem with advanced visualisation techniques like AR, the deployment of interactive dashboards and mobile applications stands as a pivotal cornerstone in fostering comprehensive engagement with the intricate world of soil digital twins. While not reliant on AR or extended reality (XR) per se, these interfaces provide a dynamic portal into the layers of data and information encapsulated within the soil ecosystem models. Dashboards, meticulously designed with data visualisation and user experience in mind, offer researchers, policy-makers, farmers, and other stakeholders an intuitive means to explore, analyse, and interpret complex soil-related data. Mobile applications further extend this accessibility, facilitating on-the-go interactions and informed decision-making for individuals across the Quadruple Helix spectrum. Through visually compelling representations, these interfaces empower users to delve into diverse dimensions of soil processes, enabling them to grasp interconnected complexities, assess trends, and strategise for sustainable land use practices. This democratisation of data-driven insights fosters a harmonious synergy between academia, industry, governance, and society, transcending conventional silos and catalysing collaborative initiatives aimed at nurturing the soil resource and its multifaceted contributions to our ecosystem.

### 3.2. Technological Stack of a Cognitive Digital Twin

The development and operation of a cognitive digital twin requires a robust technological stack, integrating various components to ensure seamless functioning and efficient data processing.

### 3.2.1. Sensor Networks and Internet of Things

Sensor networks and IoT devices play a crucial role in data collection, providing real-time information on environmental variables such as temperature, humidity, soil moisture, and air quality. These networks contribute to the continuous monitoring and updating of the cognitive digital twin. As the IoT connects a wide range of physical objects, it introduces heterogeneity by operating on a multitude of diverse devices. Therefore, there is a need for a unified architecture or middleware to effectively implement the

IoT across this diverse ecosystem [82]. Such a solution is the open-source Eclipse Ditto framework (https://eclipse.dev/ditto/, accessed on 12 September 2023), which serves as an IoT middleware, offering an abstraction layer that facilitates the interaction between IoT solutions and physical devices using the digital twin pattern. The integration of devices is achieved through device connectivity layers or MQTT brokers.

An important aspect in that regard is edge computing [83]. By processing and analysing data at or near the data source, edge computing minimises latency, enhances real-time responsiveness, and reduces the burden on network infrastructure [84]. This distributed approach optimally caters to the dynamic demands of IoT devices and applications, enabling localised decision-making, data filtering, and immediate responses. As a result, edge computing augments the efficiency, security, and scalability of IoT deployments while fostering resource efficiency. In the realm of soil monitoring, edge computing may be used from soil sensors to pre-process or compress information to near-real-time processing of the large UAV or LiDAR data to generate second-order products that are stored in the cloud (as opposed to original raw data). Thus, edge computing may be performed both by high-performance ARM-based micro-controllers that are integrated into the sensors themselves (or on hardware middleware), as well as by more-powerful CPUs.

### 3.2.2. Data Infrastructure and Management

A comprehensive data infrastructure is necessary to handle the vast amounts of heterogeneous data generated by Earth observation systems, sensors, and IoT devices. This data influx demands an intricately designed infrastructure that encompasses sophisticated data-storage mechanisms, efficient data-processing pipelines, and meticulous data-management systems. The fundamental goal of this infrastructure is to not only facilitate real-time data streams, but also to ensure the accuracy, integrity, and accessibility of the collected data. Moreover, the infrastructure must possess the capacity to store and manage the entire lifecycle data of the system.

To address these challenges, the big data infrastructure for the cognitive soil digital twin must exhibit the following essential attributes:

- Scalability and elasticity: The architecture should possess the capacity to scale seamlessly as the volume and complexity of incoming data expand. It should also exhibit elasticity to handle sudden surges in data influx without compromising performance. Various databases provide both horizontal and vertical scaling, e.g., MongoDB, DynamoDB, and Cassandra [85].
- Data integration and fusion: The data infrastructure must be capable of seamlessly integrating data from various sources, regardless of format or origin. This entails overcoming data silos and harmonising data from disparate sensors and platforms to provide a coherent and holistic view of soil behaviour. For example, platforms and initiatives such as the SensorThings API or Telegraf (https://www.influxdata.com/time-series-platform/telegraf/, accessed on 12 September 2023) provide standardised interfaces for seamless integration and management of sensor data [86].
- Real-time processing: Given the dynamic nature of soil systems and the need for timely decision-making, the architecture's data-processing capabilities should be optimised for real-time or near-real-time analysis. This empowers users to make informed decisions promptly.
- Data quality and validation: Ensuring the accuracy and reliability of data is crucial. The infrastructure should include mechanisms for data quality assessment and validation, identifying outliers or errors that could lead to misleading insights. The exact mechanisms used in each database management system differ, but most address these using data constraints, enforcing data types, using referential integrity, and/or providing trigger mechanisms (e.g., to detect outliers before inserting new data).
- Security and privacy: As data encompass sensitive information, the infrastructure must be fortified with robust security measures to safeguard against unauthorised access, data breaches, and cyber threats. Data privacy concerns, including compliance

with regulations such as the General Data Protection Regulation (GDPR), must also be rigorously addressed.

- Transparency and ownership: A clear delineation of data ownership and sharing rights is essential to establish trust among stakeholders. Additionally, transparency in data-processing and analytics methodologies fosters credibility in the insights derived from the digital twin.

Modern data lakes (either cloud-based or on-premise) comply with all of the above requirements [87,88]. A data lake serves as a centralised depository capable of accommodating extensive volumes of both structured and unstructured data (Figure 3). It facilitates the storage of raw data without the prerequisite of prior data structuring, as it can accept both structured and unstructured data. Furthermore, it extends support to diverse analytical methodologies, encompassing tasks ranging from generating dashboards and visualisations to the execution of comprehensive big data processing, real-time analytics, and machine learning, all designed to inform and enhance decision-making processes.

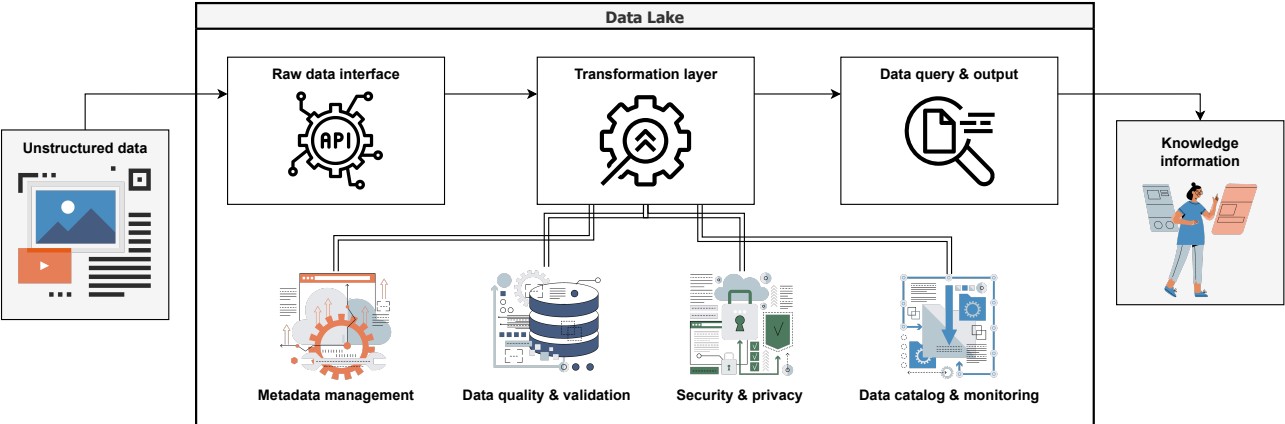

**Figure 3.** The components of a modern, scalable, data lake ecosystem acting as a centralised data repository.

3.2.3. Cloud Computing and High-Performance Computing

The computational demands of a cognitive digital twin necessitate the use of cloud computing and high-performance computing (HPC) resources: real-time performance stands as a pivotal benchmark for evaluating the efficacy of digital twins. Cloud platforms provide scalable storage and processing capabilities, while HPC systems enable intensive computational tasks required for large-scale simulations and data analysis [89]. Notwithstanding the high throughput of modern hardware, enhancing the execution speed of these systems further necessitates strategic refinements to data and algorithmic structures. In the domain of DT applications, adopting a holistic perspective is paramount—one that encompasses the comprehensive performance of the computing platform, the time delays inherent in data transmission networks, the computational prowess of cloud computing platforms, and the strategic blueprinting of an optimal system computing architecture. This architecture should be adept at fulfilling the real-time analytical and computational prerequisites of the system. It is crucial to recognise that the digital computing capabilities intrinsic to the platform hold the reins of the system's overall performance, serving as the cornerstone of its computational bedrock.

In this regard, contemporary platforms that can serve the soil digital twin may utilise the JRC big data analytics platform to generate spatially explicit indicators for large areas (https://jeodpp.jrc.ec.europa.eu/bdap/, accessed on 12 September 2023), Google Earth Engine [63], or datacube platforms [90].

### 3.2.4. AI/ML Algorithms and Models

The implementation of AI/ML algorithms and models forms a fundamental component of the technological stack. These algorithms enable data analysis, pattern recognition, predictive modelling, and decision support, enhancing the cognitive capabilities of the digital twin. A structured approach in the development and production of the models should be followed across their entire lifecycle to facilitate the automation and operationalisation thereof. The field of machine learning operations (MLOps) describes a set of best practices that should be followed, covering aspects such as CI/CD automation, workflow orchestration, reproducibility, versioning, collaboration, continuous training and evaluation, metadata tracking, monitoring, and feedback [91,92]. An example of such a framework is Kafka-ML [93], which is designed to oversee the lifecycle of ML/AI applications within production environments, all facilitated by a seamless flow of continuous data streams.

### 3.2.5. Visualisation and User Interface Tools

Digital twin implementations often rely on a variety of visualisation and user interface tools to provide interactive, immersive, and informative experiences for users. Most commonly, a dashboard is used to visualise the various data recorded. A typical tool is Grafana (https://grafana.com/, accessed on 12 September 2023), which serves as the user interface, acting as the front-end for end-users. This technology offers robust support for visualising metrics sourced from popular databases like InfluxDB. Utilising the query language specific to the chosen data source, Grafana enables dynamic querying and showcases outcomes through diverse interactive panels. These panels are integral to customizable dashboards, which can be tailored according to user preferences. Furthermore, Grafana encompasses a role-based access control system. Its notable feature lies in the creation of personalised panels and the integration of versatile functionalities using plugins, supported by comprehensive libraries and documentation. Other solutions building on web technologies and visualisation engines such as d3.js (https://d3js.org/, accessed on 12 September 2023) or plotly (https://plotly.com/, accessed on 12 September 2023) may also be used.

Typically, 3D renders are also incorporated. These can be developed using 3D visualisation platforms that are commonly found in the gaming industry, such as unity (https://unity.com/, accessed on 12 September 2023) and blender (https://blender.org/, accessed on 12 September 2023). The generated models may then be incorporated into the dashboard using WebGL technology.

Finally, with respect to AR/XR technologies that can help establish immersive visualisations, the most-widespread are ARCore (for Android) and ARKit (for iOS), which are development platforms for creating augmented reality applications. VR headsets (e.g., Oculus Rift) also provide their own development interfaces to help create immersive representations.

### 3.2.6. Backup and Disaster-Recovery Systems

Implementing resilient backup and disaster-recovery systems is paramount to guarantee the utmost integrity and accessibility of vital data and essential system components. By deploying cutting-edge automated backup solutions, bolstered by robust data replication mechanisms and fortified with redundant infrastructure, the potential for data loss and system downtime is significantly mitigated. These measures work harmoniously to create a fortified safety net that safeguards against unexpected disruptions, whether caused by hardware failures, natural disasters, or cybersecurity threats. This proactive approach ensures the continuity of operations, empowers swift data restoration, and maintains a seamless user experience, ultimately reinforcing the organisation's resilience and fortitude in the face of unforeseen challenges. Various open-source and closed-source systems offer such layers of protection. Volume and filesystem snapshots can be provided inherently by most operating systems (including Linux) and database systems (e.g., MongoDB), while various cloud providers offer cloud-based solutions for backups (e.g., Microsoft Azure Backup).

### 3.2.7. Performance Evaluation and Iterative Improvement

The performance of the cognitive digital twin should be continuously assessed through systematic comparisons between simulation outputs and real-world observations, as well as leveraging expert insights. This ongoing evaluation process serves as a compass, ensuring the accuracy and fidelity of the model's predictive capabilities. By actively engaging domain experts, stakeholders, and end-users, valuable feedback is harvested, contributing to a continuous cycle of enhancement. This iterative approach, informed by diverse perspectives, fuels a relentless pursuit of refinement, resulting in a cognitive digital twin that aligns ever-more closely with real-world dynamics. The collaborative efforts to fine-tune and optimise the model translate into a potent decision-making tool that resonates with the complexities of the actual system, ultimately amplifying its value and impact within the ecosystem it represents.

## 4. Challenges and Opportunities

The proposed cognitive soil digital twin, despite its advantages, needs to overcome some challenges in order to be effectively used. At the same time, some opportunities arise.

### 4.1. Challenges for Developing a Soil Digital Twin

#### 4.1.1. Data Acquisition and Integration

Without reliable data, it is impossible to provide a real-time dynamic mapping of the physical entity. However, gathering accurate and diverse soil data in real-time to cover the multi-faceted soil functions is challenging, as it requires a number of different sensors with enough spatial coverage to be installed or used, which need to reliably work and be monitored for failures. At the same time, integrating all these data from various sources, such as sensors, satellites, and historical records, necessitates the usage of multiple sub-components of the ingestion interface to work with the various protocols and endpoints specified by each manufacturer or data provider.

Another important aspect that needs careful consideration is the data curation and pre-processing, which primarily should take place for each data source independently, in order to deal with missing data, data inconsistencies, and other quality issues. Techniques such as gap-filling of time series or remote sensing data may be employed [94,95], while automated procedures to detect outliers can help preserve data quality [96].

In addition to the challenges of data acquisition and pre-processing, it is essential to acknowledge the limitation of error propagation when fusing data from diverse sources [97]. Each data source, whether it is ground-based sensors, satellite observations, or historical records, inherently comes with its own margin of error and uncertainty. When these datasets are integrated within the framework of the cognitive soil digital twin, these errors can accumulate and propagate, potentially leading to inaccuracies in the overall model and predictions. The fusion of data with varying error profiles demands careful consideration, as the uncertainties from one source may interact with those from another. Mitigating this limitation requires robust error modelling, propagation analysis, and quality control measures at each stage of data integration. Furthermore, to address these challenges, it is imperative to develop and implement model uncertainty estimation techniques within the cognitive soil digital twin. These techniques can help quantify the inherent uncertainties within the predictive models themselves, providing a more-comprehensive understanding of the overall accuracy and reliability of the system's outputs [98,99]. Properly accounting for model uncertainty is an essential step in enhancing the robustness of the cognitive soil digital twin and ensuring its practical applicability in real-world scenarios.

#### 4.1.2. Model Accuracy, Uncertainty, and Validation

Although AI/ML models (and, in particular, deep learning models) and process-based models have exhibited reliable accuracy in the modelling of specific soil processes or quantifying soil properties, nevertheless, they are not panacea. Developing sophisticated models that accurately represent the entire complex behaviour of soil systems and the

intricate cross-independencies between its ecosystem functions is still a challenge that needs to be addressed. Of course, this can only take place with (a) continuous and reliable monitoring, which provides the essential data to benchmark these models and improve them, and (b) more-elaborate or -appropriate models.

At the same time, the use of more complex and computationally intensive workloads is a two-edged sword, as one should always balance model accuracy with computational efficiency to ensure real-time simulations. It should also be noted that the accuracy of the digital twin's predictions should be coupled with uncertainty estimations, as they foster transparency, accountability, and responsible decision-making [100,101]. They, thus, provide a means to acknowledge the inherent unpredictability in many situations and help the users of the digital twin navigate uncertainty with greater confidence and effectiveness.

Finally, validating the accuracy of the digital twin against real-world soil conditions should take place regularly, to ensure the reliability of the models' processes. This, however, incurs a cost associated with the sampling and analytical methods that need to take place to independently verify the digital twin's outputs.

### 4.1.3. Privacy and Security

Although the use of cloud storage and computing greatly facilitates the development of the digital twin, their adoption elevates potential risks, such as unauthorised access, data leakage, disclosure of sensitive information, and breaches of privacy [102]. The use of cloud technologies increases the attack surface; ergo, implementing robust encryption, access controls, and intrusion-detection systems is essential to mitigate security risks. Security measures should be enforced to safeguard sensitive soil data and data from other sources, as well as preventing unauthorised access. Considering also that the digital twin may also involve actuation, only authorised users should have access to sensitive parts of the system; thus, access control mechanisms must be installed [103,104].

It is also important to ensure compliance with data protection regulations and take into account the ethical considerations. Some of the data used by the digital twin may be of a personal or sensitive nature, particularly when they include geospatial data and localised soil conditions. Adhering to data protection regulations such as the GDPR in Europe or similar laws globally is non-negotiable [105,106]. Failure to comply can result in severe legal consequences, necessitating meticulous data anonymisation, secure storage, and controlled access mechanisms to protect individuals' privacy and meet regulatory requirements. Ethical considerations include issues related to consent, data ownership, and the responsible handling of data to prevent misuse or harm.

### 4.1.4. End-User Engagement and Acceptance

End-user engagement and acceptance are foundational challenges in the deployment of a cognitive soil digital twin within the scientific and agricultural communities. The assimilation of novel technological paradigms can often encounter significant resistance, stemming from diverse factors, including entrenched traditional methodologies, apprehensions regarding the technological intricacies, and the learning curve associated with embracing an advanced computational tool set.

To navigate these challenges effectively, it becomes imperative to substantiate its worth through the delivery of palpable and immediate benefits. Such evidence is instrumental in dissipating initial scepticism and galvanising user engagement. These benefits may encompass, for instance, the digital twin's capacity to optimise irrigation practices, increase crop yields, reduce resource profligacy, and refine land use planning. The articulation of these advantages is not only pivotal, but also demands precision to appeal to the discerning scientific and agricultural audience. Educational initiatives form an essential component of this endeavour: facilitation mechanisms, such as didactic workshops, structured training programs, and comprehensive user manuals, underpin this effort.

Additionally, the iterative and dynamic nature of user feedback holds profound significance. The integration of end-user insights into the developmental and refinement

phases in its entire lifecycle is indispensable. It not only enriches its functionality, but also conveys a responsiveness to user exigencies and aspirations. This user-centric approach contributes substantively to the acceptance and continual utility of the digital twin.

### 4.2. Opportunities of the System

### 4.2.1. Data-Driven Research and Analysis Using Co-Design

The soil cognitive digital twin fosters the development of transdisciplinary research and innovation ecosystems and acts as an accelerator to co-create knowledge and innovations. The involvement of all actors of the Quadruple Helix is, thus, integral. Researchers and stakeholders can actively engage with a cognitive digital twin to advance scientific understanding, inform decision-making processes, and address environmental challenges. Researchers can leverage the wealth of integrated data and simulation capabilities offered by a cognitive digital twin to conduct data-driven research, analyse ecosystem dynamics, and gain insights into complex environmental phenomena.

The cognitive digital twin serves as a collaborative platform, enabling growers, researchers, institutions, industry experts, policy-makers, and other stakeholders to share data, models, and insights. This fosters interdisciplinary collaboration, encourages knowledge exchange, and facilitates collective efforts towards sustainable ecosystem management, allowing stakeholders to share best practices, lessons learned, and innovative ideas. The digital twin will serve as a virtual meeting point, enabling remote collaboration and ensuring wider participation in soil research and innovation.

### 4.2.2. Education and Outreach

One of the most-important aspects of the cognitive digital twin is that it facilitates research and education through a virtual experimentation platform for different soil scenarios. This platform empowers researchers to simulate and assess a spectrum of soil conditions, facilitating controlled experimentation in silico. This capability is particularly valuable in addressing the challenges of practical experimentation in the field, such as resource constraints, time-intensive data collection, and environmental factors. Researchers can leverage the digital twin to test hypotheses, validate models, and explore diverse soil dynamics, thereby advancing the frontier of soil science and related disciplines.

The digital twin assumes also a dual-role as an educational catalyst, supporting educational initiatives through interactive learning tools. It not only empowers practitioners and researchers, but also functions as an interactive learning tool to augment the understanding of soil processes and behaviours among students and the broader public. This educational facet aids in disseminating knowledge about soil science in a comprehensible and engaging manner. By visualising complex ecosystem processes and engaging the public through interactive interfaces, the digital twin fosters a deeper understanding of environmental issues among diverse audiences. Through these user-friendly interfaces and visualisations, the digital twin demystifies complex soil concepts and fosters a deeper appreciation of soil's role in agriculture, environmental conservation, and land management. As an educational resource, it equips learners with the insights necessary to make informed decisions and contributes to building a community of soil-aware citizens and professionals, thereby increasing soil literacy in society.

### 4.2.3. Innovation in Sensor Technology

Another noteworthy opportunity is the potential to drive significant advancements in sensor technology, by either developing new sensor systems or improving the capacities of existing systems. Soil data collection is highly dependent on the accuracy, granularity, and cost-effectiveness of sensors. The digital twin can act as a catalyst for the development of state-of-the-art soil sensors that offer improved precision and cost-efficiency. It effectively serves as a test bed for the rapid prototyping and testing of new sensor technologies. Researchers and manufacturers can collaborate to iterate and fine-tune sensor designs in real-world soil conditions. This accelerates the development cycle of cutting-edge

sensors, ultimately leading to more-accurate and -affordable data-collection tools. It can also encourage the exploration of diverse sensor modalities, including optical, chemical, biological, and remote sensing technologies.

The capabilities of the digital twin can also be continuously expanded and enhanced, through the incorporation of improved data inputs. As technology evolves, new sources of data become available (e.g., new sensors), which can be seamlessly integrated, enriching the understanding of soil behaviour and enabling more-robust predictions. Continuous improvement and adaptation can be achieved through feedback loops with users and stakeholders. Their insights and needs can drive further enhancements in the data inputs, ensuring that the digital twin remains relevant and valuable over time.

### 4.2.4. Policy Evaluation: Climate Resilience and Adaptation and Soil Health

The cognitive soil digital twin harbours substantial potential for policy evaluation, particularly in the domains of climate resilience, adaptation, and soil health. This tool offers stakeholders and policy-makers a versatile platform to comprehensively assess the potential repercussions of policy interventions, spanning diverse realms like land-use modifications, climate adaptation tactics, and pollution-mitigation measures. By simulating and visualising scenarios, the digital twin assists in evidence-based decision-making and policy formulation.

In the context of climate resilience and adaptation, the digital twin plays a pivotal role by forecasting the ramifications of climate change on soil behaviour. This foresight enables the strategic development of adaptive measures that mitigate adverse impacts and enhance resilience in the face of changing environmental conditions. Furthermore, it supports sustainable land use planning by meticulously evaluating soil responses to evolving climatic circumstances. This informs decisions regarding optimal land utilisation, fostering practices that align with both agricultural productivity and environmental sustainability objectives.

Its utility extends beyond climate considerations to encompass efficient resource management. Data-driven decision-making, facilitated by the digital twin, empowers stakeholders to minimise resource wastage and curtail environmental impacts. By providing actionable insights to farmers, land planners, and policy-makers, it equips them with a powerful tool set for navigating the complexities of soil management and land use planning. In this manner, the digital twin not only contributes to policy formulation, but also aids in the realisation of sustainable and resilient soil ecosystems, ultimately benefiting both society and the environment.

### 5. Conclusions

In this paper, we presented a comprehensive framework for the cognitive soil digital twin, ushering in a new era in soil science and ecosystem monitoring. The framework not only introduced the conceptual underpinnings, but also delineated the architectural elements and essential building blocks necessary for the realisation of this technology. Emphasis was placed on the technological stack underpinning the digital twin, underscoring its capacity to integrate vast and diverse datasets and perform the simulation of ecosystem services, thereby providing a holistic perspective on the soil ecosystem.

Moreover, this study delved into the critical aspects of both the limitations and opportunities inherent in this paradigm. We elucidated the challenges encompassing data integration and data privacy, model accuracy, and end-user engagement. Simultaneously, we underscored the immense potential of the digital twin, spanning policy evaluation, data-driven research, education and soil literacy in society, sensor development, and enhanced soil health monitoring.

As we contemplate the future, it is evident that the implementation of an open-source framework, rooted in the principles of collaboration and knowledge sharing, is imperative. Such a framework holds the potential to not only support the advancement of soil science, but also transcend disciplinary boundaries, nurturing cross-sectoral collaboration. With a commitment to transparency and open access, the framework can act as a catalyst for a new

wave of soil research, agricultural sustainability, and environmental stewardship. As soil science and related fields stand at the precipice of transformation, embracing the cognitive soil digital twin heralds a promising path toward a more-sustainable and -informed future.

**Author Contributions:** Conceptualisation, N.L.T.; methodology, N.L.T., N.S. and E.K.; writing—original draft preparation, N.L.T. and N.S.; writing—review and editing, E.K.; supervision, E.K. and G.C.Z.; funding acquisition, E.K. All authors have read and agreed to the published version of the manuscript.

**Funding:** This research received no external funding.

**Institutional Review Board Statement:** Not applicable.

**Informed Consent Statement:** Not applicable.

**Data Availability Statement:** No new data were created nor analysed in this study. Data sharing is not applicable to this article.

**Conflicts of Interest:** The authors declare no conflict of interest.

## Abbreviations

The following abbreviations are used in this manuscript:

| | |
|---|---|
| AI | Artificial intelligence |
| API | Application Programming Interface |
| AR | Augmented reality |
| CPU | Central Processing Unit |
| ESA | European Space Agency |
| GDPR | General Data Protection Regulation |
| HPC | high-performance computing |
| IoT | Internet of Things |
| MIR | Mid-infrared |
| ML | Machine learning |
| UAV | Unmanned aerial vehicle |
| VNIR | Visible to near infrared |
| XR | Extended reality |

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
