# Peer review of "Cognitive Soil Digital Twin for Monitoring the Soil Ecosystem: A Conceptual Framework"

_soilsystems, doi:10.3390/soilsystems7040088_

Round 1
Reviewer 1 Report
Digital twin is a popular concept in recent years. Soil Digital Twin acts as a bridge between scientific insights and policy implementation in the field of soil science. The paper proposed a conceptual framework and discussed the characteristics and components of the framework. The work is interesting and comprehensive. But when I look into the paper and find that there are a few points that could be possibly improved, as shown below,
1. For Figure 1. The fundamental characteristics of a Cognitive Soil Digital Twin. I noticed "System state snapshots, Data storage architectures, and Historic & health state analysis" were categaried in "Lifecycle Data management". I don't understand the logic for this grouping. Obviously "Historic & health state analysis" is at the level of analysis and why it is put in the same level as the other two and in Data management?
2. Many parts (components) in Figure 2 simply duplicate those from Figure 1. It is helpful to highlight the difference in the two figures or at least explain the linkage between them, as the two figures look very similar. For example, the physical layer in Fig. 2 corresponds to the Physical counterpart in Fig. 1, and DataStorage / Preparation in Fig. 2 looks similar to the Lifecycle Data management in Fig. 1
3. All the presented work was based on "conceptual" aspect. There is no real implementation from the practical aspect. I suggest, if possible, to provide some work how the conceptual framework was realized by presenting a prototype.
The language looks good to me.
Author Response
Dear reviewer,
many thanks for your constructive criticism. Please see the attachment for our point-by-point reply.
With best wishes,
Nikos

Reviewer 2 Report
This is a good manuscript, which reported a comprehensive framework for the Cognitive Soil Digital Twin. The authors had documented some challenges encompassing data integration and data privacy, model accuracy, and end-user engagement. At present, this manuscript is well written and can be considered for acceptance. Some minor comments are mentioned at the bottom for authors' reference.
1) Normally soil moisture and temperature and even some soil nutrients may be easily determined in the field by using soil sensors or/and satellites. However, fusion and error analysis of data from these different sources would have an important limitation on the Cognitive Soil Digital Twin. For the soil environment with a high heterogeneity, is there possible to obtain some necessary parameters for reasonably predicting soil carbon stock in terrestrial ecosystems at regional scales? If possible, the authors would give some comments and/or explanations in the text, which would be interesting to the readers.
2) Lines 152-155: the authors mentioned the model Roth-C model to predict soil carbon dynamics. Within the Roth-C model, some parameters included different soil organic C pools such as microbial biomass C, humus and inert organic matter. Can these soil organic C parameters be currently measured via soil sensors or satellites?
Author Response

(The authors gave the same response as above.)
